# Dense Cover, but Not Allelopathic Potential, of Naturalized Alien *Cenchrus echinatus* L. Threatens the Native Species in Urban Vegetation

**DOI:** 10.3390/plants12213736

**Published:** 2023-10-31

**Authors:** Mahmoud O. Hassan, Howida Y. Mohamed, Mohammad K. Okla, Bushra Hafeez Kiani, Ahmed Amro

**Affiliations:** 1Department of Botany and Microbiology, Faculty of Science, Beni-Suef University, Beni-Suef E-62511, Egypt; howidayacoup71@yahoo.com; 2Botany and Microbiology Department, College of Science, King Saud University, Riyadh 11451, Saudi Arabia; malokla@ksu.edu.sa; 3Department of Biology and Biotechnology, Worcester Polytechnic Institute, Worcester, MA 01609, USA; bushra.hafeez@iiu.edu.pk; 4Department of Botany and Microbiology, Faculty of Science, Assiut University, Assiut 71516, Egypt; ahmed.amro81@yahoo.com

**Keywords:** naturalized alien species, allelopathy, *Cenchrus echinatus*, weeds, floristic diversity, urban ecology

## Abstract

Exotic plants usually exhibit problems for native species where they coexist. This study evaluated the effect of naturalized alien *Cenchrus echinatus* L. on native plants in urban vegetation. A field trial was conducted to assess the effect of this species on the cover and diversity of the native vegetation. The allelopathic potential of such species was examined. Sites comprising *C*. *echinatus* had a lower cover than some native species. Lower floristic diversity was observed at higher densities of this plant. The soil under this plant attained lower N, P, and K contents. This soil had no effect on the germination and growth of native species. It also comprised germinable seeds of some species which were absent from the standing vegetation. Exotic *C. echinatus* may exert negative effects on the native vegetation of the urban plant communities. A dense cover of this species may inhibit the germination of native species, leading to a reduction in their cover. Reduction in cover and diversity of native species may not be attributed to allelopathy. These results suggest that naturalized *C. echinatus* may be more competitive than the native ones, particularly at higher densities. Furthermore, it may represent a threat to the native plants in the urban vegetation.

## 1. Introduction

In many parts of the world, invasion by alien plants usually represents a threat to natural ecosystems. The alien species causes a significant reduction in the resident species’ diversity, a disturbance in community structure, inhibition in vegetation growth, and pronounced modification in soil bioprocesses [1,2,3,4,5]. These alien species have been introduced from a certain region to another by human activities, either accidentally or intentionally [6]. Due to their destructive effects, scientists have to highlight these species and understand how biological invasions proceed and affect biodiversity. Consequently, it would be necessary to search for solutions to save these natural ecosystems [7]. Numerous studies have been conducted to show the effects of these alien species. Moreover, they have been increasing recently. However, most of these studies address the effect of invasive species, and little is known about the effects of naturalized ones. Therefore, it was necessary to fill this gap in knowledge.

Invasion success may be attributed to many factors that facilitate the dominance and widespread nature of these species. Allelopathy and competition are key factors contributing to the successful establishment of an alien species in the invaded communities [8]. Exotics may have a stronger competitive ability for resources than native ones, causing adverse impacts or excluding the natives from their community [9,10]. This highly competitive ability may be attributed to some performance-related traits, such as high growth rate, leaf-area allocation, size, and fitness, which promote invasion under different environmental conditions [11,12]. The dense cover of a species certainly facilitates competition of exotics. On the other hand, these species may produce allelopathic compounds that can manifest substantial inhibition for the associated plants directly by suppressing germination and growth performance, or indirectly by affecting soil microbes [8]. These compounds may be released into the soil via different methods such as exudation from plant roots, leaching from leaves, or the decomposition of plant litter [13,14]. Both mechanisms seem to play, at least, a partial role in the substantial decrease in the abundance of resident species in the invaded communities. Since naturalized species are a subcategory under the alien ones and extend into wide areas [15], they may exhibit adverse effects in the same manner as the alien species do. In this investigation, we will study the effect of the apparent cover and allelopathic potential of *Cenchrus echinatus* L., a naturalized alien species in Egypt, on the native associates. We also will examine its allelopathic potential using its rhizosphere soil which may have allelopathic activity.

Globally, urban expansion continues to grow with significant increments in population growth and human activities. Urban green areas are considered to be crucial components of newly urbanized communities that necessarily attract citizens. Urban ecology is also considered to be a field of interest for ecologists to study. Due to urbanization and plantation practices, new soils can be derived from old agricultural fields, and small drainage canals can be introduced into this region in order to be more suitable for the creation of gardens and parks. This practice may permit the dispersal of plant species to this habitat. Consequently, urban green spaces may become hot points for non-native species, and it would, therefore, be necessary to focus on this topic [16].

Non-native species are more closely associated with urban environments than residential ones [17]. This association may be due to the influx of their propagules [18], habitat heterogeneity [19], and their ability to adapt to high levels of disturbances [20]. For these reasons, the urban environment accommodates a high number of exotics that dominate a wide range of habitats and displace the natives [21]. Consequently, the presence of non-native plants in urban vegetation, in particular, may threaten the native species. Several studies emphasize the harmful effects of alien species on the associated species [22,23]. However, studies testing the effect of naturalized aliens in new urban areas are somewhat limited.

*Cenchrus echinatus* L. is an annual weed species, native to tropical America, and has been introduced as a garden weed in Egypt [6]. It has also been recorded in gardens and parks of newly urbanized cities [24,25,26]. This plant can reproduce by seeds that are spread widely within their spiny fruits. Fruits with burrs can easily attach to animals, vehicles, and clothing. They may also be dispersed by water and in contaminated agricultural products. It has a high capacity to spread and establish itself in newly reclaimed areas [27]. This species was recorded as a naturalized alien species in Egypt [6,28]. The allelopathic effect of this species has been scarcely explored. In this regard, previous studies investigated that the methanol and aqueous extracts from the *C. echinatus* shoot and root extracts can, to some extent, inhibit the seed germination and growth criteria of some weeds and crops [29,30,31]. Nevertheless, this is an insufficient baseline work in the field of ecological allelopathy. In addition, its potential negative effects, as an alien species, in different plant communities are still unknown. Therefore, it was necessary to gather more information about this issue. Our field observations indicated the presence of such species in urban gardens and parks, with a clear reduction in the cover of the native vegetation. The main hypothesis of this study was that the dense cover of *C. echinatus* in the field may inhibit germination or, at least, reduce the cover of the native species. This hypothesis could be tested in different ways. First, the covering areas and diversity of the native species were measured, in a field-based study, in selected sites comprising such species as well as locations completely free from it. Second, estimating the difference in soil–seed bank composition collected from *C. echinatus*-free and dominated sites may show an indication of the effect of this alien species. Existing germinable/viable seeds in a location dominated by the alien species and free from the species observed in the seed bank may provide proof for competition and/or allelopathy. As the density of *C. echinatus* increases, the potential for floristic diversity decreases. This can be assessed by measuring the diversity indices at different densities of this plant. In addition, the measurement of nutrient availability in the soils of these sites may reveal if this species is a strong competitor or not due to its potential nutrient withdrawal. Third, the allelopathic potential of this species can be tested using the soil collected from sites with and without *Cenchrus*, as the rhizosphere soil of this species may contain allelopathic candidates inhibiting germination and the growth of some target native species.

## 2. Results

### 2.1. Floristic Composition

In the vegetation involving *Cenchrus echinatus*, ten weed species belonging to five families were detected. Half of these species fall within the members of Poaceae, while the remaining species belong to five other families of equal contribution. Moreover, seven of these species were annuals (Table 1).

### 2.2. Effect of Cenchrus echinatus L. on Vegetation

In general, the cover of most of the native species was reduced in the *Cenchrus*-affected sites (Table 1). All annuals were significantly affected (*p* < 0.01). Amongst them, *Dactyloctenium aegyptium* seemed to be displaced completely from the regions with this weed. Moreover, *Euphorbia peplus* was markedly affected in comparison with the remaining annuals. For perennials, it was clear that the cover of the cultivated *Cynodon dactylon* significantly declined (*p* < 0.01) in the locations comprising this taxon, whilst the remaining perennials were not affected (Table 1).

With respect to the measured diversity indices, there was a remarkable reduction trend for them as the density of *Cenchrus* increases (Figure 1). The negative effect of this species on plant diversity varied concerning the measured index and the cover value of *C. echinatus*. In more detail, both species richness and Simpson index began to reduce at the cover of *Cenchrus* ≥ 50%, while both evenness and Shannon–Wiener indices declined at *Cenchrus* cover ≥ 60% (Figure 1). Moreover, all of these indices were inversely correlated with the cover of the alien *C. echinatus* (Table 2).

Regarding the investigation of the soil seed bank in the sites with *C. echinatus*, four species were found to be emerging (Table 3). These species were absent in the *Cenchrus*-free sites. The highest numbers of seeds were recorded for *Dactyloctenium aegyptium*, after which the number of germinating seeds of both *Amaranthus viridis* and *Euphorbia peplus* were statistically similar. Fewer numbers of seeds were detected for *C. echinatus* itself (Table 3).

### 2.3. Effect of C. echinatus on Soil Properties

In terms of soil criteria, the parts of the urban vegetation with *Cenchrus* attained significantly lower contents of available macronutrients; namely nitrogen, phosphorus, and potassium. However, these locations had higher amounts of available zinc (Table 4). The remaining soil properties were statistically similar in all studied sites.

### 2.4. Allelopathic Potential of C. echinatus

The results indicated that the soil under such exotic species had no effect on the germination and growth of the three target native species (Figure 2).

## 3. Discussion

The present investigation showed that the patches of the gardens and parks where the alien *Cenchrus echinatus* was recorded attained a lower cover of the native species. This is, maybe, one of the more important and interesting discoveries of this study. This observation was similar to that obtained by Hassan and Mohamed [3] who explained the negative effect of naturalized alien *Paspalum dilatatum* on the native urban vegetation. What we found in this study was the existence of viable/germinable seeds of different native species in the soil under *C*. *echinatus*. It is also worth mentioning that those species were completely absent in the above-ground vegetation in the locations where their seeds were found. This result substantially reflects the negative effect of the dense cover of the exotic plant on the seed germination of the native ones. Consequently, the reduction in cover of the native species may be related to decreased seed germination. On the other hand, the measured floristic diversity in this work was density-dependent. In addition, the negative correlation between the cover of *C. echinatus* and the measured diversity indices was significant. In this regard, it was reported that the interference effect of *C*. *echinatus* was higher as the density of this weed increased [32]. Germination may be delayed or completely inhibited due to the interspecific neighborhoods [33]. Moreover, the negative density-dependent germination was observed by several authors, e.g., [34,35].

Allelopathy may have a role in weed interference and declining species diversity [36]. Moreover, naturalized alien plants may exert an allelopathic effect that inversely affects the cover of the associated species [3,37]. Nascimento et al. [29] showed that tissues of the root and shoot systems of the alien *C. echinatus* contained volatile allelochemicals (mainly palmitic, oleic, and stearic acids, and methyl linolelaidate). However, the present work showed that the collected soils from areas occupied by *C*. *echinatus* had a non-significant effect on seed germination and the growth performance of selected target species. Nascimento et al. [29] noticed that shoot extract was more effective than that of root extract. This may reflect the weaker activity of root exudates. In addition, the volatilization potential of these allelochemicals may reduce their activity after soil collection and the conducting of a test under greenhouse conditions for some time without the donor plant. This result indicates that the effect of alien *C. echinatus* may be related to its existence under field conditions. This notion may align with the results obtained by Hassan et al. [35] who found that the living *Sonchus oleraceus* had more adverse effects on the target species in comparison with its rhizosphere soil-only equivalent. This result may also confirm that the effect of *C. echinatus* due to its allelopathic potential was uncertain, and an additional interaction other than allelopathy may be involved.

Our findings showed that some species were more affected by the coexistence of the alien species in comparison with others. In particular, the results indicated that *Dactyloctenium agyptium* was completely displaced from the sites involving *C. echinatus*. In addition, *Euphorbia peplus* suffered greatly. This effect could be attributed to the rarity of these species. It was shown that rarer species were subjected to more suppression by the community dominants than the introduced grasses [38]. In addition, Zhang and van Kleunen [9] concluded that common naturalized alien species were more competitive than rare native species, and this might lead to the loss of rare native species. Furthermore, our field observations indicated the tendency of the alien species to prevail, resulting in more inhibition for the native ones. The existence of both native species in the seed bank suggests the inhibitory effect of *C. echinatus* on the germination of these species. The complete displacement of *D*. *aegyptium* may be related to the complete inhibition of its seed germination exerted by the alien species under field conditions. Its existence in the seed bank, but its absence in the standing vegetation, suggests the inability of this species, in particular, to cope with conditions of the above-ground vegetation [39], perhaps exerted by the dense cover of the alien species.

One of the most remarkable results was the effect of *C. echinatus* on soil properties. This study revealed that the sites comprising this species were of lower soil macronutrients, particularly the available N, P, and K. Such results could reflect the heavy removal of these nutrients made by *C. echinatus* over the native species. This effect may explain why the cover and diversity of native vegetation declined. Nutrient deficiency may lead to a reduction in overall biodiversity [40]. Furthermore, high plant diversity could be associated with high levels of soil macronutrients [41,42]. On the other hand, this result may also show the better competitive ability of such alien species compared with the native plants. This connotation may be in agreement with that obtained by Tecco et al. [43] who showed that a superior competitive ability of alien species over that of natives is often associated with a high ability to acquire and retain resources. Moreover, these plants may show stronger competitive ability over native ones [44].

## 4. Materials and Methods

### 4.1. Vegetation Survey

A field study was conducted in the study area described by Hassan [24] in 2018 and repeated during the 2021 growing seasons. The average climatic conditions always indicate a hot dry summer and a temperate, slightly rainy winter. The average meteorological data of the study area in 2018 were reported by Hassan and Hassan [25], and those that were monitored in 2021 are illustrated in Table 5. A total of 80 plots, each of 10 × 1 m, were randomly sampled from the vegetation representing the gardens and parks in the study area. Amongst them, 40 plots were chosen in the sites with *C. echinatus* (the invaded plots), whereas the remaining plots were selected in the *Cenchrus*-free sites (control plots). On average, the control plots were selected about 5 m apart from those comprising *C. echinatus* in order to share similar environmental conditions. The total number of plots selected was proportional to the area occupied by this alien weed. Plot size was selected on the basis of a similar study performed in this area [25]. In each plot, the names of all species determined were listed. Species identification and nomenclature were carried out using Boulos [45,46,47,48]. The covering area of the species (expressed as m^2^) was virtually determined, and the relative cover of each species (pi) (expressed as %) was calculated through the equation: relative cover = [cover of a species i/cover of all species] × 100. In addition, the floristic diversity was measured as four diversity indices: species richness (S), i.e., the total number of plant species observed, Shannon–Wiener index (H′), Evenness index (E), and Simpson’s index (D) [49,50] as follows:H′ = −∑(pi × ln pi)
E = [−∑(pi × ln pi)]/ln S
D = 1/C and C = ∑pi^2^

### 4.2. Soil Seed Bank Sampling

The soil seed bank was sampled in the stands representing the dense cover of *C*. *echinatus* (cover of *Cenchrus* ≥ 70% and the remaining vegetation was covered by the cultivated *Cynodon*) to avoid the existence of seeds of other species from the above-ground vegetation, and to ascertain that the inhibition of the germination of the detected seeds could be attributed to the presence of the alien species. At such cover %, a reduction in plant diversity was highly apparent. Four soil samples, each of 10 cm × 10 cm at 5 cm depth, were randomly collected from the upper 5 cm of soil around representative individuals from the stands with the alien plant, while control soils were selected from the stands without this plant. This depth was considered as most seeds have a high tendency to be located at this depth [51]. Soil samples were sieved through a 2 mm sieve to remove plant litter. Known volumes of sieved soil samples were spread randomly in a 2 cm layer overlying sterilized soil in 10 cm × 12 cm pots in a greenhouse sharing the prevailing environmental conditions at Beni-Suef University and regularly irrigated. Control pots contained the soils free from *C. echinatus* for a potential comparison with the seeds detected in *Cenchrus*-infested soils. The germinable/viable seeds were assessed by monitoring the emergence of the seedlings. Emerging seedlings were counted and left to be juvenile for possible identification by the mentioned Boulos handbooks.

### 4.3. Soil Analysis

From each plot, three soil samples were collected from 0–20 cm depth and pooled together forming one composite. They were air-dried, sieved through a 2 mm sieve, and stored in plastic bags for analysis. Determination of soil texture, pH, electrical conductivity (EC), and the contents of CaCO_3_, organic carbon (OC), and availability of some nutrients including N, P, K, and Zn was performed.

For the determination of soil texture, i.e., the percentage of sand, silt, and clay in the inorganic fraction of the soil, the hydrometer method was followed [52]. Both pH and EC were measured in soil–water extract using deionized water. Soil pH was measured in the soil extract (1:2.5 *w*/*v*) with a digital pH meter (AD 3000, Adwa, Szeged, Hungary), while EC was measured in another soil extract (1:5 *w*/*v*) using a conductivity meter (Jenway 3305, Jenway, Stone, UK). The soil content of CaCO_3_ was estimated using the titration method [53]. Soil organic carbon was measured using the Walkley–Black method [54]. The available soil nutrients including nitrogen, phosphorus, potassium, zinc, and copper in the soil samples were determined using Allen’s methods [55].

### 4.4. Allelopathic Potential of C. echinatus

#### 4.4.1. Soil Sampling and Preparation

Soil samples from the rhizosphere of *C. echinatus* in *Cenchrus*-dominating sites were collected and exposed to effective shaking inside a plastic bag to obtain one composite as the treatment soil (the invaded soil). The non-rhizosphere soil samples (control soil) were collected from the sites dominated by Bermuda grass (*Cynodon dactylon*); the native species cultivated in these gardens and parks for greening purposes. Both soil types were sieved (2 mm size) to remove the plant litter and placed in plastic pots (10 cm diameter × 15 cm depth each) for near sowing of the target species. Each pot contained about one kg of air-dried soil.

#### 4.4.2. The Target Species

Three native target species, namely: *Amaranthus viridis* (Linn.), *Dactyloctenium aegyptium* (L.) Willd., and *Eragrostis pilosa* (L.) P.Beauv. were used as test species. These species are common weeds in the study area [25]. The preliminary field observation showed that *D. aegyptium* appeared to have been away/apart from the sites with *C. echinatus*, whereas the remaining species had a lower cover in these sites in comparison with the *Cenchrus*-free areas. This observation may reflect the impact level of the alien species. Ripe seeds of these species were collected from different locations constituting the gardens and parks of the study area (i.e., the New Beni-Suef city) at the late fruiting stage.

#### 4.4.3. Test for Allelopathy

Thirty seeds of the target species were equally disseminated at 0.3 cm depth of soil per pot. The pots were irrigated regularly via a misting process when needed. The experiment was retained in a protected site under the prevailing environmental conditions in a completely randomized design with four replicates for 30 days. At the end of the test, the number of emerging seedlings was determined in order to calculate the germination percentage. Growth criteria including shoot height, root depth, and the total biomass were measured per each individual as indicated by Hassan and Mohamed [3]. The shoot length of each individual was measured from the apical bud to the base of the stem, while that of the roots was determined as the length of the main root. To determine the biomass, the emerging individuals were oven-dried at 70 °C for 72 h. The masses of such specimens were weighed to the nearest milligram per pot. The leaf area per each weed species was measured by weighing their tracings on a high-quality paper and comparing them with a paper of known area and weight.

### 4.5. Statistical Analyses

The data were first checked for normality and homogeneity of variances using the Kolmogorov–Smirnoff and Levene’s tests, respectively. The results meeting the requirement of normality and homoscedasticity were analyzed using the independent samples *t*-test. When field data were not normally distributed, the non-parametric Mann–Whitney U test was performed. To evaluate the relationship between the cover of the exotic *C. echinatus* and the different diversity indices, a Pearson correlation analysis was carried out. All analyses were carried out using SPSS, version 20.0 (IBM Corporation, Armonk, NY, USA).

## 5. Conclusions

To a great extent, the results of this study followed the main hypothesis proposed. The coexistence of the exotic naturalized *Cenchrus echinatus* in the urban vegetation led to a remarkable decline in the cover of most of the associated native plants. Rare species; particularly *Dactyloctenium aegyptium* and *Euphorbia peplus*, were more affected in comparison with the remaining native species. Reduction in the measured diversity indices was density-dependent. Moreover, all of these indices were inversely correlated with the cover values of such plants. Interestingly, the presence of this alien species resulted in a significant reduction in the available soil macronutrients including nitrogen, phosphorus, and potassium. In addition, the soil under this species indicated the existence of germinable/viable seeds of some species which were absent from the standing vegetation. With respect to the allelopathic potential of this exotic species, its rhizosphere soil had no effect on the germination and growth performance of some selected native plants under greenhouse conditions, which may reflect another possible interaction from *C. echinatus* in the urban vegetation. The reduction in cover and diversity of the resident plants may be attributed to the reduction of soil-available macronutrients made by *C. echinatus*. Moreover, the failure of germination in some native species in the field may be related to the dense cover of the exotic plant. Consequently, the naturalized alien *C. echinatus* may be more competitive than the native ones. The allelopathic potential of such species seemed to be an uncertain mechanism to exhibit the obtained reduction in cover and diversity of native species. Till now, *C. echinatus* has not been recorded as invasive. However, its negative effects on the native vegetation and soil were addressed. This study substantially indicated that naturalized alien *C. echinatus* represents a threat to the urban vegetation. Conservation strategies should be devoted to controlling alien species in urban ecosystems.

## Figures and Tables

**Figure 1 plants-12-03736-f001:**
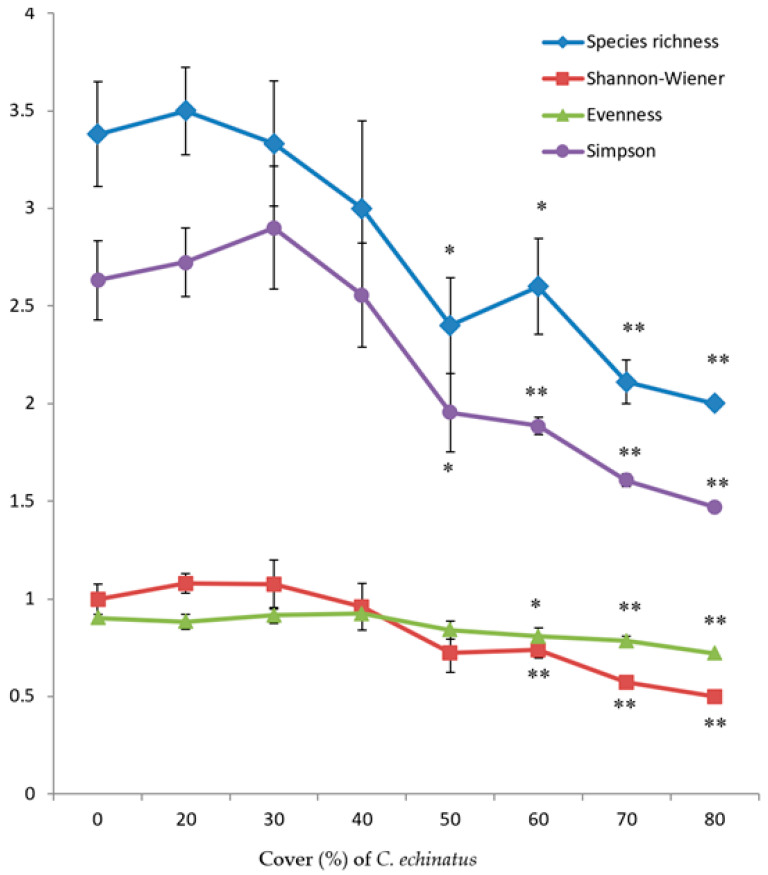
Measured diversity indices (Mean ± SE) in relation to the different densities (%) of *C. echinatus*. Density at 0 value represents *Cenchrus*-free stands. * Significant result from *Cenchrus*-free stands at *p* < 0.05. ** Significant result from *Cenchrus*-free stands at *p* < 0.01.

**Figure 2 plants-12-03736-f002:**
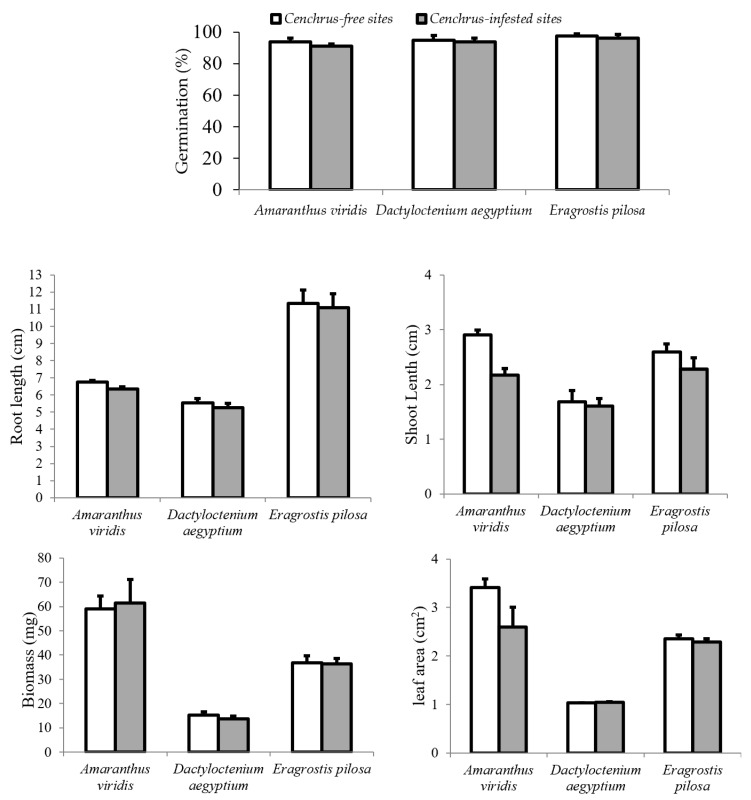
Germination (%) and the measured growth criteria (Mean, bars represent standard errors) of the selected target species associated with *C. echinatus* when planted in *Cenchrus*-infested and *Cenchrus*-free soils.

**Table 1 plants-12-03736-t001:** Mean coverage values (m^2^) (mean ± SE) of the plant species detected in the *Cenchrus*-free and *Cenchrus*-infested sites.

Species	Family	*Cenchrus*-Free Sites	*Cenchrus*-Infested Sites
*Amaranthus viridis* L. †	Amaranthaceae	39.82 ± 4.23	15.10 ** ± 4.41
*Apium leptophyllym* Pers. †	Apiaceae	15.60 ± 3.48	3.9 ** ± 0.45
*Cynodon dactylon* Pers. ††	Poaceae	142.21 ± 13.25	97.38 ** ± 9.46
*Dactyloctenium aegyptium* (L.) Willd. †	Poaceae	3.30 ± 0.64	0.0 ** ± 0.0
*Dichanthium annulatum* (Forssk.) Stapf ††	Poaceae	19.28 ± 4.48	8.50 ± 3.86
*Digitaria sanguinalis* (L.) Scop. †	Poaceae	16.38 ± 3.18	11.90 ± 3.27
*Eragrostis pilosa* (L.) P.Beauv. †	Poaceae	49.74 ± 8.42	14.81 ** ± 3.50
*Euphorbia peplus* L. †	Euphorbiaceae	4.35 ± 0.60	0.35 ** ± 0.26
*Oxalis corniculata* L. ††	Oxalidaceae	17.11 ± 6.79	9.44 ± 2.33
*Plantago lagopus* L. †	Plantaginaceae	17.00 ± 2.54	3.63 ** ± 2.63

** *p* < 0.01, † Annual, †† Perennial.

**Table 2 plants-12-03736-t002:** Correlation coefficients (r) between the measured diversity indices and the cover of *C. echinatus* L.

Diversity Indices	Cover of *C. echinatus*
Species richness	−0.54 **
Shannon index	−0.70 **
Evenness	−0.35 *
Simpson index	−0.71 **

* *p* < 0.05, ** *p* < 0.01.

**Table 3 plants-12-03736-t003:** Number of germinating seeds (Mean ± SE) of the species per pot detected in the seed bank in the soil dominated by *C. echinatus* compared with the *Cenchrus*-free sites.

Species	Life Span	*Cenchrus*-Free Sites	*Cenchrus*-Infested Sites
*Amaranthus viridis* L.	Annual	-	9.72 ** ± 1.01 b
*Cenchrus echinatus* L.	Annual	-	2.73 ** ± 0.33 a
*Dactyloctenium aegyptium* (L.) Willd.	Annual	-	14.80 ** ± 0.90 c
*Euphorbia peplus* L.	Annual	-	9.64 ** ± 0.98 b

- Not detected. ** significant results from the *Cenchrus*-free sites at *p* < 0.01. Values sharing different letters within a column are significant results among the species detected in the *Cenchrus*-infested sites at *p* < 0.05.

**Table 4 plants-12-03736-t004:** Soil physicochemical properties (mean ± SE) in the *Cenchrus*-free and *Cenchrus*-infested sites.

Soil Properties	*Cenchrus*-Free Sites	*Cenchrus*-Infested Sites
Sand (%)	48.39 ± 2.15	49.00 ± 1.60
Silt (%)	32.62 ±1.58	32.02 ± 1.50
Clay (%)	18.99 ± 0.93	18.78 ± 0.70
pH	7.98 ± 0.03	7.92 ± 0.03
EC (µS cm^−1^)	377.54 ± 15.50	377.08 ± 10.50
CaCO_3_ (%)	2.47 ± 0.09	2.56 ± 0.07
OC (%)	1.97 ± 0.07	2.05 ± 0.05
Available nutrients (mg kg^−1^ soil)		
N	67.2 ± 1.50	51.12 ** ± 3.90
P	12.55 ± 0.42	10.50 ** ± 0.45
K	411.02 ± 5.70	361.15 ** ± 6.20
Zn	1.60 ± 0.13	6.10 ** ± 0.31
Cu	2.28 ± 0.13	2.45 ± 0.14

** *p* < 0.01.

**Table 5 plants-12-03736-t005:** The average meteorological data of the study area during the 2021 growing season. The climate data were obtained from the Agrometeorological Application Research Department (AARD), the Central Laboratory for Agricultural Climate (CLAC), Egypt.

Climatic Parameters	Jan.	Feb.	Mar.	Apr.	May.	Jun.	Jul.	Aug.	Sep.	Oct.	Nov.	Dec.
Maximum temperature (°C)	29	28	36	41	44	42	43	44	40	36	35	24
Minimum temperature (°C)	3	2	5	5	16	16	20	21	18	14	11	3
Relative humidity (%)	52.25	52.19	49.5	36.75	27.31	33.38	32.62	33.19	43.75	48.38	52.31	62.19
Precipitation (mm)	0	0	10.54	0.30	0	0	0	0	0	0.30	4.50	9.3
Wind speed (km h^−1^)	31.36	35.96	35.64	33.62	33.48	25.45	24.70	28.73	34.34	27.32	31.42	35.57

## Data Availability

The data presented in this study are available on request from the corresponding author.

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
