# Peer review of "Dense Cover, but Not Allelopathic Potential, of Naturalized Alien *Cenchrus echinatus* L. Threatens the Native Species in Urban Vegetation"

_plants, 2023, doi:10.3390/plants12213736_

Round 1

Reviewer 1 Report

Comments and Suggestions for Authors

The effect of Cenchrus echinatus L. on the germination, growth and vegetal cover of native urban species is evaluated. The authors demonstrate the negative impact of the exotic plant on the native plants.

The work is well introduced, the reader is adequately contextualized, the knowledge gap is well defined and the research objective is well defined. On the other hand, the methodology is well described and supported. 

1. The results are well described, however the discussion could be improved. 

a) What are the environmental conditions of the study area.

b) What are the morphological characteristics of the autochthonous species, in order to explain the possible relationship between morphology and percentage of affectation of each autochthonous species by the presence of the exotic species. 

c) The authors indicate that allelopathy can affect the diversity of autochthonous species, but there is no relationship with possible secondary metabolites. Talking about allelopathy makes it necessary to relate possible secondary metabolites.  

d) Justify in a better way the effect of the exotic species on shoot length, biomass and leaf area, in relation to the allelopathic effect. These three factors are usually related to light conditions. 

2. In the conclusions the authors express: "Species richness was not affected at all". Support this assertion, since the results possibly indicate the opposite.

3. Conclusions: in relation to allelopathy, the conclusions are related to results from other authors, therefore, the results of the study presented should be analyzed.

Author Response

The work is well introduced, the reader is adequately contextualized, the knowledge gap is well defined and the research objective is well defined. On the other hand, the methodology is well described and supported. 

  1. The results are well described, however the discussion could be improved. 

Discussion was improved as possible in the light of new data due to statistics.

  1. a) What are the environmental conditions of the study area.

The meteorological data of the study area during the work were inserted in a new TABLE.

  1. b) What are the morphological characteristics of the autochthonous species, in order to explain the possible relationship between morphology and percentage of affectation of each autochthonous species by the presence of the exotic species. 

Really, the idea is interesting. I have done a quick search on scopus about the relation between the degree of effect of exotics on the native species with respect to their morphology. I have not find such an enough relation. What I can say about morphology is the degree of similarity between the exotic Cenchrus echinatus and many of the associated weeds as they are graminaceous, with suitable environmental condition either from climate or soil. This may enhance naturalization of this species. But, we couldn't an enough prove to infer the effect of such exotic on a certain species depending on its morphology.  

  1. c) The authors indicate that allelopathy can affect the diversity of autochthonous species, but there is no relationship with possible secondary metabolites. Talking about allelopathy makes it necessary to relate possible secondary metabolites. 

Yes, you are right. We know that allelopathic activity is related to secondary metabolites. We, therefore, inserted a short statement about it from previous studies on Cenchrus in section Discussion.

  1. d) Justify in a better way the effect of the exotic species on shoot length, biomass and leaf area, in relation to the allelopathic effect. These three factors are usually related to light conditions. 

If you are talking about the effect of the exotic species on the associated species in terms of these criteria under field conditions, I'm so sorry if I tell you I could't measure them now as we are in autumn and the exotic species as well as the associate ones are in decline. However, I promise we will per in mind these criteria when measuring field criteria. I can tell you that there were plant species were as tall as Cenchrus. However, they were significantly affected and vice versa. I therefore may tell you that the matter may not be related to light.

If you are talking about the effect of the exotic species on the selected target species under greenhouse conditions, the light, as well as all environmental conditions, was equally distributed for all pots.

Methodology of these criteria was inserted in materials and methods (section 4.4.3).

  1. In the conclusions the authors express: "Species richness was not affected at all". Support this assertion, since the results possibly indicate the opposite.

Visually, you of course can infer that species richness was decreasing where the cover of the alien species increase. I myself wondered when I saw the results of the statistical analyses. Despite the apparent reduction in species richness, the stats. showed the opposite.

However, we repeated the stats. and presented the results in different way more interesting to the readers. You can see the figure after modification.

Again, many thanks for this point.

  1. Conclusions: in relation to allelopathy, the conclusions are related to results from other authors, therefore, the results of the study presented should be analyzed.

OK! Done.

Reviewer 2 Report

Comments and Suggestions for Authors

The presented study covers research on the invasive potential of alien Cenchrus echinatus L. species to the urban vegetation. Research of alien and invasive plants is very important to solve the issues of biodiversity uncertainties. By investigating the invasive potential of Cenchrus echinatus L., this study not only enhances our understanding of its impact on urban ecosystems but also aids in formulating effective strategies for managing and preserving indigenous flora. The introduction section of the manuscript could be rearranged in a more fluent and consistent manner. Several typing mistakes should be corrected. Lines 332-335: Please explain all the letters in the equation.

Author Response

The presented study covers research on the invasive potential of alien Cenchrus echinatus L. species to the urban vegetation. Research of alien and invasive plants is very important to solve the issues of biodiversity uncertainties. By investigating the invasive potential of Cenchrus echinatus L., this study not only enhances our understanding of its impact on urban ecosystems but also aids in formulating effective strategies for managing and preserving indigenous flora. The introduction section of the manuscript could be rearranged in a more fluent and consistent manner. Several typing mistakes should be corrected. Lines 332-335: Please explain all the letters in the equation.

 reply: Letters in the equations were mentioned above the equations. Introduction was mostly adjusted. Typing mistakes were corrected.

Reviewer 3 Report

Comments and Suggestions for Authors

The information in the manuscript must be improved.

More information in graphics/figures must be added.

Author Response

Dear Reviewer,

                       Many thanks for your kindness. Many thanks for your effort also. We modified or deleted the statements you highlighted. Introduction was adjusted and improved as possible. Conclusion was modified due to the results presented.

Reviewer 4 Report

Comments and Suggestions for Authors

The study “Dense cover, but not allelopathic potential, of naturalised alien Cenchrus echinatus L. threatens the native species in the urban vegetation” assessed the C. echinatus effect on cover and diversity of the native vegetation, with focus on the allelopathic potential. It was possible to evidence that C. echinatus may be more competitor than the native ones, specially at higher densities. In my opinion it has merit to be published in the current journal; however, I have some suggestions to improve it.

 Abstract:

It is unnecessary to make mention to “plots”. Please double check this comment for the full paper, as in the lines 178 and 198. The comparisons are not between the plots, but between the treatments.

Introduction:

It is well written, included the hypothesis that has been stated in a good way. It provides sufficient background and include all relevant references. In addition, all cited references are relevant to the research.

 Results:

The results are clearly presented. I just recommend avoiding the use of the isolated genus name “Cenchus”, please use the binomial nomenclature rule (C. echinatus).

Discussion:

Please use C. echinatus instead of Cenchrus echinatus. Double check this comment for the full paper.

 Material and Methods:

The research design is appropriate, and the methods are adequately described.

Final comment: I recommend the current paper for publication it presents originality and novelty.

Author Response

 Abstract:

It is unnecessary to make mention to “plots”. Please double check this comment for the full paper, as in the lines 178 and 198. The comparisons are not between the plots, but between the treatments.

Reply: We replaced this word with other words in most of the manuscript. However, we maintained such word in materials and methods for more details in methodology. 

Introduction:

It is well written, included the hypothesis that has been stated in a good way. It provides sufficient background and include all relevant references. In addition, all cited references are relevant to the research.

 Results:

The results are clearly presented. I just recommend avoiding the use of the isolated genus name “Cenchus”, please use the binomial nomenclature rule (C. echinatus).

Reply: we replaced the genus name with binomial name in most cases. However, we maintained a few for variation in data presentation. This is better for readers. 

Discussion:

Please use C. echinatus instead of Cenchrus echinatus. Double check this comment for the full paper.

OK! Done.

Round 2

Reviewer 1 Report

Comments and Suggestions for Authors

The authors have made the suggested changes

Reviewer 3 Report

Comments and Suggestions for Authors

You may improve the results' presentation.